# PulseImpute: A Novel Benchmark Task for Pulsative Physiological Signal Imputation

**Maxwell A. Xu[1], Alexander Moreno[1*, 2], Supriya Nagesh[1], V. Burak Aydemir[1],**
**David W. Wetter[3], Santosh Kumar[4], James M. Rehg[1]**
[1] Georgia Tech, [2] Luminous Computing, [3] University of Utah, [4] University of Memphis
{maxxu,...,rehg}@gatech.edu

## Abstract

The promise of mobile health (mHealth) is the ability to use wearable sensors to monitor participant physiology at high frequencies during daily life to enable temporally-precise health interventions. However, a major challenge is frequent missing data. Despite a rich imputation literature, existing techniques are ineffective for the pulsative signals which comprise many mHealth applications, and a lack of available datasets has stymied progress. We address this gap with *PulseImpute*, the first large-scale pulsative signal imputation challenge which includes realistic mHealth missingness models, an extensive set of baselines, and clinically-relevant downstream tasks. Our baseline models include a novel transformer-based architecture designed to exploit the structure of pulsative signals. We hope that PulseImpute will enable the ML community to tackle this important and challenging task.

## 1 Introduction

The goal of mobile health (mHealth) is to use continuously collected signals from wearable devices, such as smart watches, to passively monitor a user's health states during daily life and deliver interventions to improve health outcomes. The use of devices such as Fitbit [17] to monitor physical activity has become an established practice, with large-scale consumer adoption. Even more exciting is the increasing feasibility of measuring complex health states, such as stress [25], by leveraging high-frequency physiological signals from wearable sensing technologies. A subset of these physiological signals are *pulsative*, which we define as signals that have a quasiperiodic structure with specific signal morphologies (e.g. the QRS complex in electrocardiography (ECG)), which vary over time and across populations due to their origins in the cardiopulmonary system. ECG and Photoplethysmography (PPG) signals are examples of such pulsative signals. The rich signal structure, especially in terms of shape and timing has significant clinical value, for tasks such as heart disease diagnosis [59].

However, a key challenge is addressing *missing data*, which is commonplace and arises from multiple causes such as insecure sensor attachment or data transmission loss [47]. Current mHealth systems either employ simple imputation methods, such as KNN [44], or simply do not trigger health interventions when inputs are missing [51]. Since mHealth biomarkers may require multiple signals as input [25], the latter approach can lead to long intervals of missingness due to the juxtaposition of missingness patterns across the inputs. However, the quasiperiodic nature of these signals provides rich information for imputation, which can be exploited by modeling morphological structures over time. Additionally, the accuracy with which a signal's morphology can be recovered has a direct impact on downstream task performance, as specific waveform shapes have clinical significance and may imply specific disease states. Furthermore, the well-defined signal morphologies allow for straightforward interpretation of reconstruction. Thus, pulsative signals provide a novel context for the development of ML imputation methods, especially in comparison to prior imputation tasks.

---

*Work was done while the author was at Georgia Tech.

36th Conference on Neural Information Processing Systems (NeurIPS 2022) Track on Datasets and Benchmarks.

| | *Our PulseImpute Challenge* | *Standard Imputation Datasets [54, 26]* | *Imputation for mHealth Data Cleaning [27, 51]* | *Clinical Pulsative Signal Imputation [67, 4]* |
|---|---|---|---|---|
| mHealth Pulsative Signals | ✓ | | ✓ | |
| Publicly Available Data | ✓ | ✓ | | ✓ |
| Realistic Missingness | ✓ | ✓ | ✓ | |
| Directly Evaluates Imputation | ✓ | ✓ | | ✓ |
| Comprehensive Benchmarks | ✓ | ✓ | | |
| Downstream Tasks | ✓ | ✓ | ✓ | ✓ |

Table 1: Necessary components for an mHealth pulsative signal imputation challenge. Our PulseImpute Challenge is the only work to meet all six criteria.

We introduce *PulseImpute*, a novel pulsative signal imputation challenge to catalyze and enable the ML community to address the important missing data problems underlying current and future mHealth applications. Table 1 describes six criteria that the PulseImpute challenge provides, which no prior works address in full. We extract real missingness patterns from real-world mHealth field studies [9, 50] and mimic specific mHealth missingness paradigms [47] in order to apply these patterns to open source pulsative signal datasets. As a result, we can simulate realistic missingness while using the original ablated samples as ground truth, making it possible to quantify and visualize the accuracy of imputation. We also include three clinical downstream tasks, Heartbeat Detection in ECG, Heartbeat Detection in PPG, and Cardiac Pathophysiology Multi-label Classification in ECG, which allow us precisely interpret reconstructions in the context of their clinical utility. PulseImpute features an extensive benchmark suite of imputation methods, covering both traditional and deep-learning-based approaches. In particular, we introduce a novel transformer baseline with a Bottleneck Dilated Convolution self-attention module that is designed for the pulsative signal structure and provides state-of-the-art (SOTA) performance. These baselines provide a strong context for future research efforts.

To summarize, we make the contribution of introducing the PulseImpute Challenge, which is composed of 1) a comprehensive benchmark suite for mHealth pulsative signal imputation with publicly-available data across multiple signal modalities and reproducible missingness models; 2) nine baseline models which demonstrate the failure of existing time-series imputation methods to address our novel challenge; and 3) an additional novel baseline incorporating a self-attention module which learns to attend to quasiperiodic features and delivers SOTA performance. The challenge code and datasets can be found at `www.github.com/rehg-lab/pulseimpute` and `www.doi.org/10.5281/zenodo.7129965`, respectively.

## 2   Related Work

Prior works are summarized in Table 1 and include: 1) Standard time-series imputation datasets and associated deep-learning methods; 2) Imputation approaches in mHealth data cleaning and preprocessing; and 3) Clinical pulsative signal imputation research. PulseImpute provides the first comprehensive imputation benchmark for mHealth pulsative signals.

**Standard Imputation Datasets**: Prior time-series imputation work uses Traffic [15], Air Quality [26], Billiard Ball Trajectory [18], Sales [2], and other miscellaneous time-series modalities. Health-related imputation works [53, 16, 66, 36, 10] have benchmarked clinical record imputation [54, 29] with non-pulsative signals (e.g. hourly body temperature). All of these prior datasets lack *the high frequency and variable yet specific morphologies of pulsative waveforms* [25, 27]. For context, the 5-minute, 100 Hz waveforms in our curated dataset are ∼600 times longer than the time-series data found in the clinical records dataset, PhysioNet 2012 [54].

Furthermore, many of these works [7, 40] simulate missingness by dropping independent time points at random, which is *not representative of real-world mHealth missingness patterns*, as visualized in Figure 1 and Appendix A3. State-of-the-art deep-learning time-series imputation methods [7, 2, 38] that were developed in these settings perform poorly in our PulseImpute Challenge (see Section 5).

**Imputation for mHealth Data Cleaning**: [51] develops a stress biomarker with mHealth ECG field data that handles real-world missingness. However, they do not quantify the effect of missingness

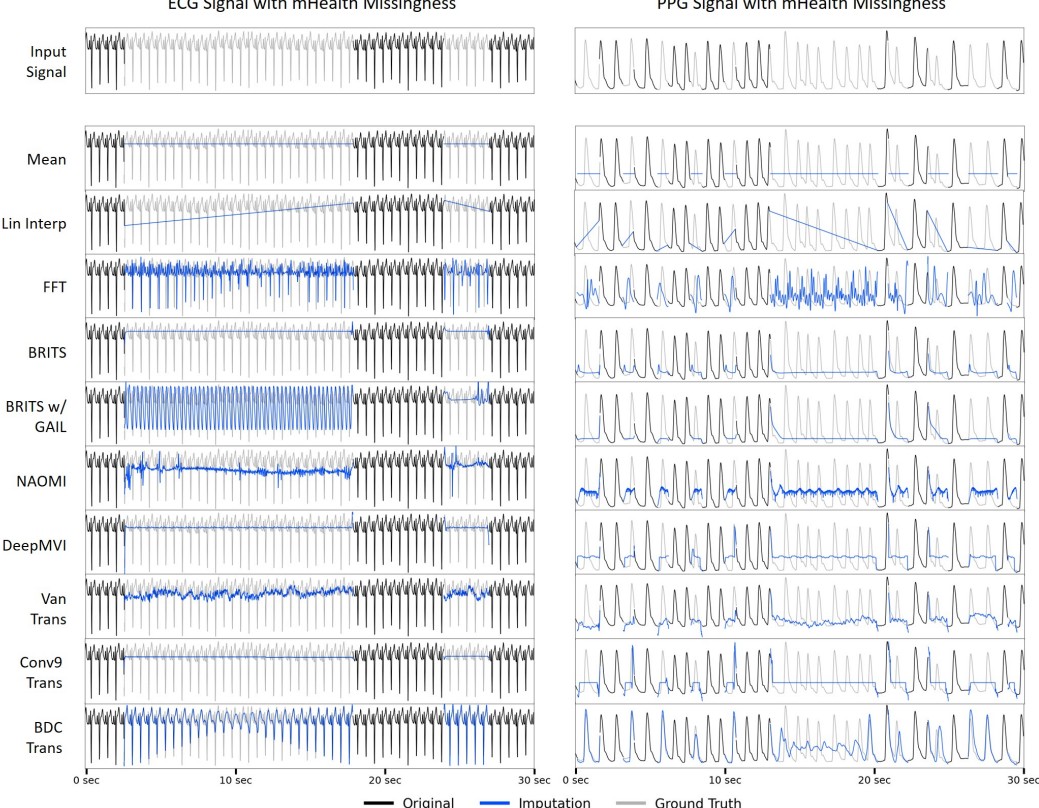

Figure 1: Visualization of imputation results on ECG and PPG signals. The large gaps found in real-world missingness patterns create substantial challenges for all methods. BRITS and DeepMVI produce nearly constant outputs. GAN-based approaches (BRITS w/ GAIL and NAOMI) hallucinate incorrect structures in incorrect locations. Our BDC Transformer also struggles in the middle of long gaps, but has the strongest performance overall, most closely reconstructing the ground-truth.

and use a simple KNN multivariate imputation method. Their dataset is not publicly-available and lacks ground truth imputation values, so it cannot be used as a benchmark. [27] also addresses stress detection in a private dataset and only benchmarks simple methods to handle missingness: a multivariate iterative imputer, mean filling, last observation carried forward, or a removal of instances with missingness. [44] utilizes KNN multivariate imputation in an accelerometry dataset, but these signals are not pulsative. These representative works illustrate how current mHealth imputation research has not yet advanced to utilize modern deep learning methods, instead treating imputation as a data preprocessing problem. PulseImpute fills a unique niche to help advance the mHealth field.

**Clinical Pulsative Signals Imputation**: Prior work has focused on using non-deep-learning imputation methods [4, 67, 42] on pulsative signals to address *multi-channel* ECG imputation on publicly-available datasets. The key distinction is that these multi-channel methods borrow information across ECG channels for imputation, a straight-forward task since the channels are highly-correlated (each channel measures the heart's electrical activity with respect to a different measurement axis). Multi-channel ECG recordings are routinely captured in clinical settings, but, in mHealth applications, where ECG is continuously measured with a wearable sensor, single-channel recording is the only practical approach. Therefore, we focus on *single-channel* imputation. This requires the much more interesting and challenging task of borrowing information across time for imputation rather than across highly-correlated channels. Please see Appendix A1.2 for further discussion.

## 3 PulseImpute Challenge Description

We focus PulseImpute on the imputation of ECG and PPG signals, illustrated in Figure 1, because these widely-available pulsative signals are used in a wide range of mHealth and clinical tasks, such

as monitoring atrial fibrillation [23], vascular aging [8], respiration rate [21], and stress [25].[1] To quantify imputation performance, we simulate missing data by ablating samples, so that the imputed samples can be compared to the original ground-truth values. Prior imputation datasets have used relatively simple approaches to simulate missingness, typically by independently removing samples at random [7, 40, 38]. We adopt two approaches for generating the block missingness patterns that characterize the mHealth domain. The first is by extracting patterns of missingness from real-world mHealth studies [9, 50], illustrated in Figure 1 and Appendix A3. The second is by randomly selecting samples with a fixed amount of missingness for ablation. Across experiments, we can vary the amount of missingness to quantify the impact of the amount of missingness on algorithm performance. We impute each signal modality independently (i.e. univariate time series imputation) because this is important for the univariate mHealth systems and leads to a more interesting and challenging task.

We generate training and testing sets in the PulseImpute Challenge by applying our missingness models to waveform data obtained from two existing clinical datasets: MIMIC-III Waveforms [43] (containing ECG and PPG signals) and PTB-XL [62] (containing ECG signals). These datasets are large-scale, freely-available, and support a variety of downstream tasks for quantifying the impact of imputation performance on derived health markers. Specifically, MIMIC-III Waveforms supports *heartbeat detection* using both ECG and PPG signals, in which the goal is to localize individual heart beats. This is a core capability that supports a variety of widely-used mHealth markers such as heart rate variability [52]. In the case of PTB-XL, the ECG waveforms have associated classification labels that fall within specific *rhythm, form, and diagnosis* categories, (e.g. aFib, inverted T-waves, WPW syndrome, respectively), which are determined at the waveform level. These comprise a complex set of downstream clinical tasks which will be directly influenced by imputation quality.

In summary, PulseImpute enables the evaluation of imputation algorithm performance at the signal level (each sample's reconstruction accuracy) and the downstream task level (quantifying the degradation in task accuracy due to imputation performance) for two widely-used pulsative signal types, ECG and PPG. In the next subsections, we describe the curation of our challenge datasets, our missingness models, and the performance metrics for our three downstream tasks: heartbeat detection with ECG, heartbeat detection with PPG, and cardiac classification with ECG. In Section 4, we describe our suite of benchmark imputation methods. In Section 5, we present empirical results that quantify the performance of SOTA methods on our novel challenge task and highlight directions for future research enabled by PulseImpute.

### 3.1 ECG Imputation and Heartbeat Detection

The goal of this task is to apply extracted real-world missingness patterns to ECG waveforms and formulate a downstream task of heartbeat detection. Imputation performance is assessed with reconstruction accuracy (signal level) and accuracy in detecting and localizing ECG peaks corresponding to the heartbeats (task level).

**Dataset**: We have curated the largest clean public ECG waveform dataset available, containing 440,953 100 Hz 5-minute ECG waveforms from 32,930 patients. Our starting point was the raw ECG signals from the 6.7 TB MIMIC-III Waveforms dataset.[2] This dataset contains a variety of waveform data (e.g. ECG, PPG, etc.) and up to three ECG leads per patient. MIMIC's unstructured nature with variable lead availability and imprecise electrode placements per recording [19] lends itself to a "union of leads" approach to dataset curation, in which we take each of the lead channels and add them separately to the dataset as individual univariate time series. This modifies the multivariate time-series into multiple univariate time-series. As previously discussed, univariate, single-channel recording is the norm for mHealth applications, and the inclusion of different leads in the union of leads dataset forces imputation methods to learn to borrow information across time to capture morphology due to each lead's distinctive shape. Additionally, while the most popular mHealth lead is lead I [1, 20], a wide range of lead configurations have been experimented with [32, 55], so this approach is useful for developing an imputation model that can generalize to different leads.

The key curation step was to filter out waveforms that were too noisy to support beat detection, while preserving those with irregular heart beat patterns. We used Welch's method [64] and identified

---

[1]We note that pulsative signals can arise in both mHealth and clinical settings, as continuous waveform data may be captured in the ICU. A more detailed comparison of these settings is in Appendix A1.

[2]MIMIC-III Waveform is a different dataset from the MIMIC-III Clinical. MIMIC-III Clinical is more commonly used and only contains low frequency vitals data, not raw waveforms.

peaks in the periodogram which reveal the harmonics of the QRS complex. Tests on the peak distribution and spacing were used to identify clean ECG signals corresponding to typical as well as abnormal heart rhythms, while rejecting noisy samples. Further details can be found in Appendix A2.1. In contrast to our approach, prior works with MIMIC-III Waveform have used a random subset (1,000 2-minute ECG signals from 50 patients [3]) of the data or a smaller matched subset that has corresponding clinical data (30,124 5-second ECG signals from 15,062 patients [31]). We believe we are the first to preprocess the ECG MIMIC-III Waveform dataset in its entirety.

The resulting curated signals are very long, measuring 30,000 time points, which adds a level of complexity towards this challenge to have an emphasis on models that are scalable. For example, a transformer's self-attention mechanism is $O(n^2)$, and therefore, cannot be applied naively. From an application's perspective, such long recordings are used because heart rate variability is most commonly measured in 5-minute intervals [52].

**Missingness**: We obtained *extracted ECG mHealth missingness* patterns from our mHealth field study with 169 participants [9] (See Appendix A2.3 for details). The missingness patterns are variable: most (69%) of the missingness gaps are 3-9 seconds long, but some (2%) of the gaps last more than a minute. Appendix A3.1 contains visualizations and further descriptions of the extracted missingness patterns. We have extracted 102,201 5-minute missingness masks, which capture the complex, real-life missingness patterns produced by wearable sensors in field conditions.

**Downstream Task**: We use the Stationary Wavelet Transform peak detector [30] to identify the sequence of peaks corresponding to individual heartbeats. Ground truth peaks are found by running the detector on the non-ablated signals. Peaks in the imputed signal are matched to the true peaks with a 50 ms tolerance window [46] to identify true vs false positives and define a detection problem (see Appendix A2.4 for details). We use the standard peak detection measures, F1 score, precision, and recall, in order to quantify performance [37, 6, 45]. 95% confidence intervals are generated by bootstrapping test samples with 1,000 iterations.

### 3.2 PPG Imputation and Heartbeat Detection

Analogous to Section 3.1, the goal of this task is to apply extracted real-world missingness patterns to PPG waveform data and formulate a downstream task of heartbeat detection in PPG.

**Dataset**: We have curated the largest clean public PPG waveform dataset available, containing 151,738 100 Hz 5-minute waveforms from 18,210 patients. We started with the raw PPG signals from MIMIC-III Waveforms, comprising of a variable-length univariate time-series for each participant cropped to be 5 minutes. To identify clean PPG waveforms and reject noisy signals, we used the approach from [61] to perform beat segmentation and an ensemble averaging approach to identify a beat template for each waveform, which then is used to obtain a per-beat quality measure. Clean recordings had 95% of the beats with quality higher than 0.5. Please see Appendix A2.2 for details.

**Missingness**: We used a real-world mHealth PPG dataset, PPG-DaLiA [50], to obtain *extracted PPG mHealth missingness* patterns. See Appendix A2.3 for details and A3.2 for visualizations of the patterns. We extracted 425 missingness masks which are used to ablate the curated PPG waveforms.

**Downstream Task**: Analogous to ECG, we used peak detection in PPG to identify individual beats (see Appendix A2.4 for details). Peaks in the clean waveforms provided ground truth for evaluating the imputed signals with F1 score, precision, and recall used to measure performance with 95% confidence intervals generated from 1,000 bootstrapped iterations.

### 3.3 ECG Imputation and Cardiac Pathophysiology Multi-label Classification

This task focuses on quantifying the downstream impact of imputation on the challenging task of classifying cardiac disease conditions from ECG signals, by systematically assessing how varying percentages of missingness impacts downstream performance.

**Dataset**: We utilize PTB-XL [62], which is composed of 21,837 100 Hz 10-second ECG waveforms from 18,885 patients, annotated with 71 labels that cover diagnostic, form, and rhythm categories of cardiac conditions. For each of the categories, the labels within them are not disjoint, resulting in a multi-label classification problem. We need to adapt the 12-lead ECG PTB-XL data and the SOTA multivariate xResNet1d classifier [57] to the univariate setting in order to create a downstream task.

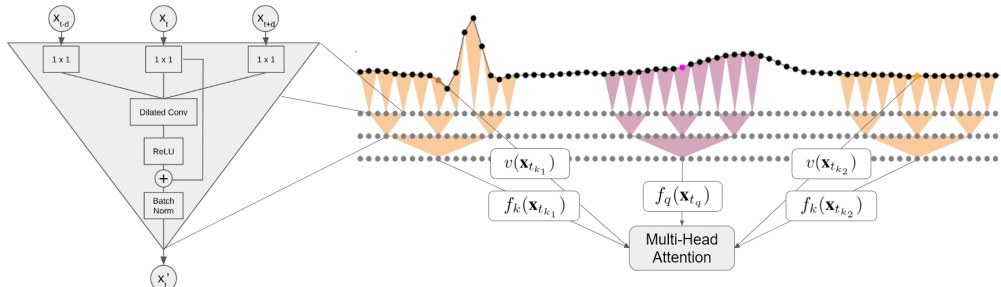

Figure 2: Novel Bottleneck Dilated Convolutional (BDC) Self-Attention Architecture. The sequence of black dots on the right illustrates the ECG time series input to the module. Each triangle on the right denotes a BDC block, shown in an expanded view on the left. The magenta dot denotes a query point for self-attention, while the orange dots denote representative queries. The query/key functions are composed of stacked blocks, denoted by the hierarchical structure of the colored triangles which illustrate the exponentially increasing dilation factor and receptive field used in the query/key functions. This enables efficient comparison of local context comprising 100s of samples.

Since our goal is to assess imputation performance on classification, the domain shifts associated with using different leads in a "union of leads" approach would be a confounding factor for the classifier. Our experiment design therefore uses one specific lead, Lead I, the most common mHealth lead [1, 20], and modifies the classifier to have an input channel size of 1.

**Missingness**: We simulate two types of missingness patterns corresponding to long and short intervals of data loss: extended missingness and transient missingness. *Extended Missingness* ablates a random single continuous set of samples as a percentage of the waveform duration. This models the most common source of missingness, sensor attachment issues, which comprise ∼85% of total missingness [48]. In contrast, *Transient Missingness* models the sporadic loss of packets of samples due to communication failures or throttling of the data collection app [48]. It is modeled by dividing the waveform into disjoint 50 ms blocks and sampling independently according to a fixed percentage of missingness to select blocks for ablation. The 50 ms block size was selected to match standard packet sizes in mHealth [25, 63]. Extended and transient missingness are both parameterized by a missingness percentage that controls (on average) the proportion of ablated samples in a waveform. In contrast to Section 3.1, our goal is to characterize the impact of varying the percentage of missing data points at test time on the reconstruction accuracy and downstream performance. Imputation models are trained at a fixed 30% missingness percentage (30% was most common amount of missingness found in a 10 sec signal in our mHealth field study [9]). During testing, samples are ablated using percentages from 10% to 50% at a step size of 10%, making it possible to quantify the effectiveness of imputation methods in generalizing to varying amounts of missingness at test time.

**Downstream Task**: With [57], three xResNet1d [22] multi-label classification models for predicting diagnosis (e.g. WPW Syndrome), form (e.g. inverted T-waves), or rhythm (e.g. aFib) labels were trained on non-ablated data. Then, for all of the extended and transient missingness scenarios, after imputing a separate, held-out dataset with a given imputation method, each of the trained xResNet1d were evaluated on the imputed waveform to quantify the impact of imputation on clinical tasks that leverage the timings and morphologies of the ECG waveforms. Classification results were measured using Macro-AUC, which is a common measure for multi-label classification under label imbalance [56, 11, 28] and has been theoretically proven to be optimized when the instance-wise margin is maximized [65]. The confidence intervals are generated identically as previously described.

## 4 Benchmarks and Proposed Bottleneck Dilated Convolutional Self-Attention

This is the first comprehensive study of mHealth pulsative signal imputation, and therefore there is a lack of prior baselines. We cover a range of classical methods to demonstrate baseline performance and use SOTA deep-learning methods from the general time-series imputation literature. In total, our benchmark suite includes the ten methods listed in Table 4 with performance shown in Section 5.

Classical methods include mean filling, linear interpolation, and a Fast Fourier Transformer (FFT) imputer. Mean filling and linear interpolation are commonly used in mHealth [34, 14] and help

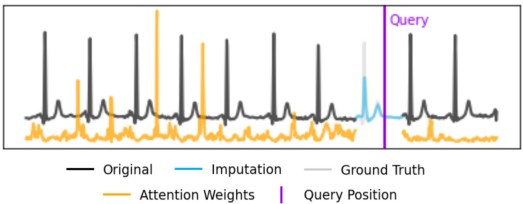

Figure 3: Visualization of attention weights of BDC self-attention for a given query position. Instead of encoding a bias for time points close to the query, as done in prior work [39, 7], BDC attends to locations which are far from the query but similar in morphology, exploiting the quasiperiodicity.

baseline the performance of more complex methods. We include an FFT imputer as a method that is able to utilize frequency information to exploit the quasiperiodic nature of the data [49].

For deep-learning methods, we include DeepMVI [2], NAOMI [38], BRITS w/ GAIL [38], BRITS [7], Vanilla Transformer [60], and Conv9 Transformer [35]. DeepMVI is a transformer-based architecture that achieves SOTA in ten diverse real-world time-series datasets, ranging from air quality to sales [2]. NAOMI develops a non-autoregressive approach paired with Generative Adversarial Imitation Learning (GAIL) [24], and achieves SOTA for imputation tasks framed around trajectory modeling [38]. NAOMI's backbone architecture is BRITS, a widely-used pure RNN imputation benchmark with a time-delayed loss propagation, that achieved SOTA in its benchmarked datasets [7]. BERT [13] used the vanilla transformer [60] with a masked language modeling imputation task for learning language representations, and the Conv9 Transformer [35] was proposed to address the lack of local context while modeling time-series, which we will further discuss below.

Most of these methods were not designed to exploit the quasiperiodicity of our pulsative signals, so we anticipated that each of these methods would perform poorly in our setting. Therefore, we develop a transformer-based architecture that can provide a SOTA baseline for this domain. We claim that the pair-wise comparisons in the transformer's self-attention module are an attractive method for modeling the quasiperiodic dependency structure of pulsative signals. In order to fully realize the potential of transformers in this setting, we identify three challenges that must be addressed: 1) local context, 2) permutation equivariance, and 3) quadratic complexity. This then motivates the development of the Bottleneck Dilated Convolutional (BDC) transformer baseline illustrated in Figure 2. We now describe how our BDC Transformer addresses the three challenges.

**Local Context:** Transformers utilize self-attention, which we define below for a given query $\mathbf{x}_q$:

$$A(\mathbf{x}_q) = \sum_{t_k \in S_{t_k}} \frac{\exp(\langle f_q(\mathbf{x}_{t_q}), f_k(\mathbf{x}_{t_k}) \rangle)}{\sum_{t'_k \in S_{t_k}} \exp(\langle f_q(\mathbf{x}_{t_q}), f_k(\mathbf{x}_{t'_k}) \rangle)} \mathbf{x}_{t_k} \mathbf{W}_v \qquad (1)$$

where $f_q(\mathbf{x}_{t_q}) = \mathbf{x}_{t_q} \mathbf{W}_q$, $f_k(\mathbf{x}_{t_k}) = \mathbf{x}_{t_k} \mathbf{W}_k$, and $\mathbf{W}_{\{q/k/v\}} \in \mathbb{R}^{D_x \times D}$ (scaling factor omitted for brevity). $S_{t_k}$ is the set of all time points that are keys, and $\mathbf{x}_t$ is the $t$th row of $\mathbf{X} \in \mathbb{R}^{T \times D_x}$, where $T$ and $D_x$ are the time-series length and dimensionality, respectively. In NLP applications, each input word token has intrinsic semantic information which allows for meaningful direct comparison via self-attention in Eq. 1. In contrast, in order to meaningfully compare two timepoints in a time-series, it is necessary to utilize the local signal context around the queries and keys. This can be accomplished via convolutional self-attention [35], which models the query and key function as a convolution, which we demonstrate for $f_q$ below:

$$f_q(\mathbf{x}_{t_q}) = (\mathbf{X} \star h)[t_q] = \sum_{s=-\infty}^{\infty} \mathbf{x}_s h_{t_q+s} \text{ where } h_u = \begin{cases} \mathbf{W}_u^{(q)} & \lfloor \frac{1-i}{2} \rfloor \leq u \leq \lfloor \frac{i-1}{2} \rfloor \\ 0 & \text{elsewhere} \end{cases}, \qquad (2)$$

where $i > 1$ is the filter size and with $\mathbf{W}_u^{(q)} \in \mathbb{R}^{D_x \times D}$ as the $u$th row of $\mathbf{W}^{(q)} \in \mathbb{R}^{i \times D_x \times D}$. The original conv transformer implementation [35] had a single convolution with a small filter size of 9. The key for this approach to be effective in our pulsative signal setting is to achieve a sufficiently large receptive field (RF) to achieve subsequence comparisons between patterns lasting for 100s of time points, while maintaining computational efficiency. We use stacked dilated convolutions with bottleneck 1x1 layers in our novel BDC architecture illustrated in Figure 2. The bottleneck reduces dimensionality, allowing us to stack filters with exponentially increasing dilation factors, thereby exponentially and efficiently increasing the RF. We can see empirically in Table 2, that while both

| Models w/ Params fixed at ~2.6 mil | Van Trans RF=1 | Conv Trans RF=9 | **BDC Trans RF=883** |
|---|---|---|---|
| MSE ↓ | 0.0177 | 0.0231 | **0.0123** |

Table 2: Comparison of models with parameters fixed at ~2.6 mil. With its stacked dilated convolutions, BDC is able to efficiently increase RF and improve performance relative to vanilla self-attention.

| Model | **BDC Trans** | PE+BDC Trans | Conv Trans | PE+Conv Trans |
|---|---|---|---|---|
| MSE ↓ | **0.0118** | 0.0121 | 0.0223 | 0.0225 |

Table 3: Positional encoding (PE) slightly degrades performance for both BDC and Conv Transformer.

BDC and conv transformers expand receptive field (RF), only BDC improves performance relative to the vanilla self-attention after controlling for parameter count.

Prior time-series imputation architectures such as BRITS [7], encode a bias for time points that are close to the query. This is not effective for pulsative signals, where temporal locations which are far from the query can be similar in morphology, and thus useful for imputation. Our BDC Transformer is able to exploit this, as shown by the learned attention weights illustrated in Figure 3.

**Permutation Equivariance:** A permutation of a transformer's inputs results in a corresponding permutation of its outputs without any change in values [68]. This is addressed in the original transformer's formulation via an additive positional encoding [60], but this does not have a good inductive bias in our setting. The absolute position relative to the start of a pulsative signal is not meaningful, due to the arbitrary start-time at which sensors begin recording [52]. Even if the signals were initially aligned, due to within-subject and between-subject phase variance stemming from the heart rate variance phenomena [52], the relative position of specific waveform shapes will vary.

Now, from Eq. 2, one can see that conv and BDC self-attention are no longer permutation equivariant because the calculation at each position depends on its neighbors. An additional additive positional encoding would perturb the original signal, potentially rendering the imputation task more difficult. Indeed, we empirically demonstrate in Table 3 that including an additive positional encoding [60] degrades model performance. Therefore, we design our approach around the BDC self-attention without a positional encoding, because of its strong inductive bias in its ability to encode local context, while also breaking permutation equivariance.

**Quadratic Complexity:** Transformers have quadratic time and space complexity for self-attention that limits applications to long sequences [58]. The Longformer [5] dilated sliding window attention restricts the key range in self-attention, $S_{t_k}$, without modifying the query/key functions, allowing it to be easily combined with our BDC self-attention. We use this longformer variant for the transformer models in the 5-minute-long (30,000 time points) time-series used in the heartbeat detection tasks.

## 5   Results

We now present comprehensive results for our baseline models on all of the PulseImpute Challenge tasks, organized by downstream task as described in Section 3. See Appendix A4 and our code repository on implementation details and reproducibility.

**ECG Imputation and Heartbeat Detection:** All prior time-series imputation models perform poorly on the long ECG time-series (30,000 time points) with complex ECG missingness patterns, which can be seen in Table 4 and the ECG column in Figure 1. BRITS w/ GAIL and NAOMI hallucinate realistic ECG patterns but do not match the ground-truth. BRITS fails to effectively impute over longer gaps. FFT and our BDC model have the best imputation performance and can reconstruct the rhythm of the missing ECG peaks, as shown in the peak detection statistics, with BDC easily having the best MSE and F1 score overall, at 0.0194 and 0.64, respectively. BDC transformer can effectively capture the extended local context in comparison to other transformer models (e.g. DeepMVI, Conv9, Vanilla), reconstructing realistic ECG signals, reminiscent of the ground-truth. However, as seen in the further visualizations in Appendix A5.1, none of the models are able to effectively impute over the extra-long missingness gaps that can last up to one minute.

**PPG Imputation and Heartbeat Detection:** PPG is morphologically simpler than ECG (see Figure 1), and most methods perform better with respect to their F1 score. However, as seen in Figure 1, methods such as DeepMVI and BRITS can only impute values near observed data. FFT and BDC

| Models | ECG Imputation and Heartbeat Detection | | | | PPG Imputation and Heartbeat Detection | | | |
|---|---|---|---|---|---|---|---|---|
| | ↓ MSE | ↑ F1 | ↑ Prec | ↑ Sens | ↓ MSE | ↑ F1 | ↑ Prec | ↑ Sens |
| Mean Filling | .0278 ± .00019 | .01 ± .000 | .60 ± .000 | .00 ± .007 | .0971 ± .00123 | NaN | NaN | 0 ± 0 |
| Lin Interp | .0467 ± .00046 | .01 ± .000 | .62 ± .000 | .00 ± .009 | .1393 ± .00073 | NaN | NaN | 0 ± 0 |
| FFT [49] | .0350 ± .00024 | .09 ± .001 | .07 ± .001 | .16 ± .001 | .1449 ± .00120 | .10 ± .001 | .07 ± .001 | .16 ± .001 |
| BRITS [7] | .0445 ± .00068 | .01 ± .000 | .32 ± .000 | .01 ± .001 | .1064 ± .00068 | .01 ± .000 | .03 ± .000 | .01 ± .001 |
| BRITS w/ GAIL [24] | .0571 ± .00068 | .05 ± .001 | .08 ± .001 | .03 ± .003 | .1102 ± .00068 | .07 ± .001 | .29 ± .001 | .04 ± .003 |
| NAOMI [38] | .0392 ± .00061 | .05 ± .001 | .13 ± .001 | .03 ± .001 | .0856 ± .00061 | .09 ± .001 | .09 ± .001 | .10 ± .001 |
| DeepMVI [2] | .0276 ± .00019 | .05 ± .000 | .49 ± .000 | .02 ± .005 | .0802 ± .00061 | .25 ± .001 | .31 ± .001 | .21 ± .002 |
| Vanilla Trans [60] | .0368 ± .00021 | .02 ± .001 | .29 ± .000 | .01 ± .005 | .0967 ± .00065 | .13 ± .001 | .15 ± .001 | .12 ± .001 |
| Conv9 Trans [35] | .0299 ± .00021 | .01 ± .000 | .47 ± .000 | .00 ± .007 | .0805 ± .00056 | .20 ± .001 | .20 ± .001 | .19 ± .001 |
| BDC Trans (ours) | **.0194** ± .00017 | **.64** ± .003 | **.83** ± .003 | **.52** ± .002 | **.0137** ± .00020 | **.81** ± .003 | **.79** ± .003 | **.83** ± .003 |

Table 4: ECG and PPG imputation and heartbeat detection results using extracted mHealth missingness patterns (see Sec. 3.1 and 3.2). Measures are for reconstruction performance (MSE) and Heartbeat Detection accuracy (F1 Score, Precision, and Sensitivity) with 95% Confidence Intervals.

demonstrate that exploiting quasiperiodicity is useful across signal modalities, and BDC achieves the best overall results as seen in Table 4. Further visualizations can be found in Appendix A5.2.

**ECG Imputation and Cardiac Classification:** The imputation visualizations and the downstream results in Figure 4 show that in the transient missingness setting, many models outperform mean imputation in mimicking the Rhythm, Form, and Diagnosis features present in the original waveform, as reflected by their downstream Macro-AUC results for each category. Here missingness gaps are shorter and learning long-term dependencies is less necessary. BRITS and Vanilla Transformer, imputation methods that were originally trained and designed for very short missingness gaps [7, 13], perform well in this setting. FFT imputation performs poorly in this setting, but our BDC model has the best MSE and does the best in reconstructing rhythm, form, and diagnosis characteristics.

In the more challenging extended missingness setting, imputation performance drops across all models, resulting in poor downstream performance. The GAN-based methods and FFT have poor performance, while the BDC model has the best MSE and the best performance in reconstructing rhythm and form features. However, in the diagnostic category, performance is poor. We hypothesize that this may be tied to BDC transformer reconstructing much shorter R peaks in the ECG signal under high missingness percentages, which can be seen in Figure A13 in the Appendix. Some diagnostic labels are dependent on R peak height (e.g. LAFB is diagnosed with tall R waves [33]), and thus will be adversely affected. We hypothesize that the minimalistic imputations produced by BRITS, Conv9, and mean filling (see Figures 3, A13) fare better because there is less misleading signal information present, suggesting that the current imputation SOTA is inadequate for this challenging task. Full tabulated results with confidence intervals and extra visualizations can be found in Appendix A5.3.

## 6  Discussion

**Future Work and Limitations:** Our BDC architecture demonstrates strong performance in exploiting quasiperiodicity across different signal modalities and missingness patterns, so future work should be done to utilize our model in a variety of pulsative signal and missingness settings (e.g. noisy ECG in fMRI settings, seismocardiogram data corrupted with motion artifacts, etc.). However, the visualizations in Figure 1 (and in Appendix A5), demonstrate that all existing methods are unable to impute over missingness gaps lasting up to a minute in the heartbeat detection tasks. Additionally, all methods are far from the upper-limit of performance in each label group for Cardiac Classification in the extended missingness setting. A key challenge is to improve imputation in the middle of long gaps, which might benefit from a generative modeling approach.

A potential benefit of mHealth is the ability to analyze an individual's health-related behaviors so as to improve their health outcomes. Future work could include benchmarking for personalized models, similar to the approach proposed by [12]. Additionally, none of our benchmarked methods explicitly model the imputation uncertainty, and we plan to explore related architectures with uncertainty modeling such as [53]. Another potential area is explainability of such imputation models.

We note that the theoretical missingness model for this work is MCAR, as the missingness is independent of the waveforms that they are applied to [41]. This was necessary to obtain ground-truth imputation targets, but future work should investigate the inclusion of MAR and MNAR missingness.

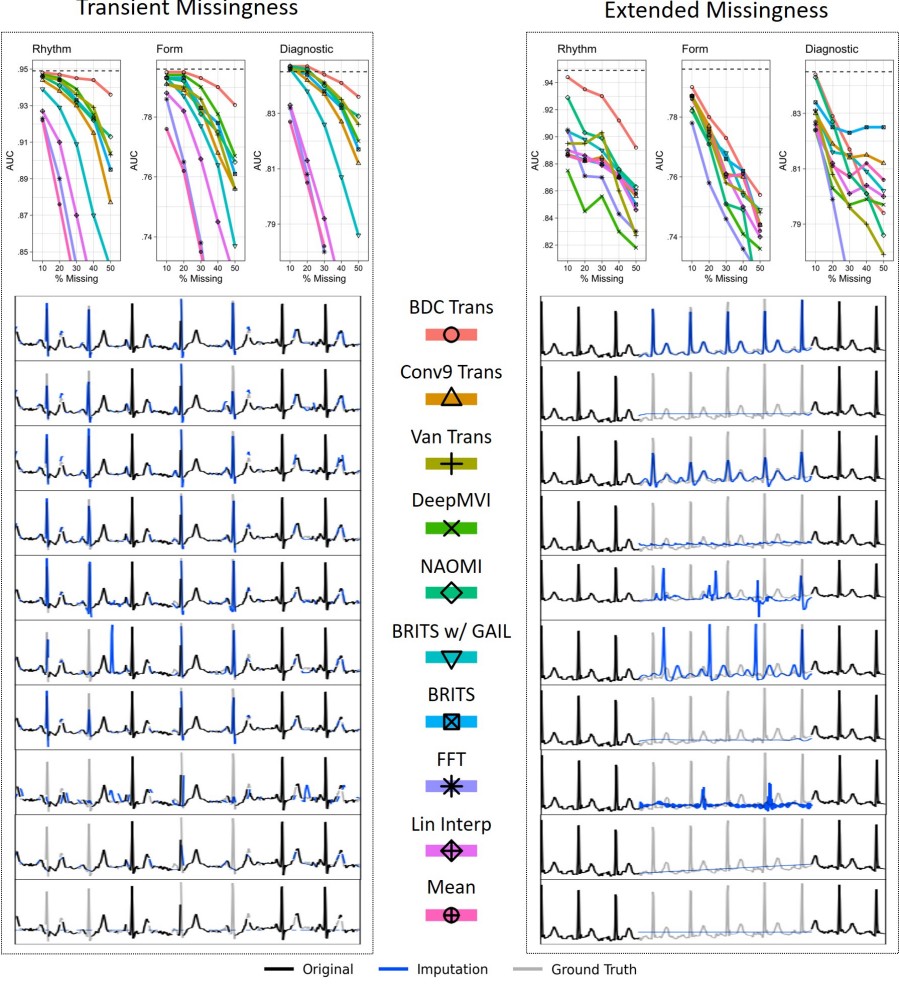

Figure 4: Cardiac Classification in ECG Results for Transient and Extended Missingness on Rhythm, Form, and Diagnosis label groups. For each label category, a cardiac classifier was trained and tested on complete data (top, illustrated by dashed line). The trained model was then evaluated on imputed test data (for five levels of missingness from 10% to 50%) produced by each baseline, yielding the Macro-AUC curves (top). Six seconds of representative imputation results for the 30% missingness test case are plotted (below). Extended Missingness proved to be more challenging for all models.

**Societal Impacts**: We anticipate our work to have positive societal benefits by enabling researchers to address one of the most common issues found in mHealth, accelerating the field forward to enable individuals to live healthier lives. As with all ML challenges, there may be a negative environmental impact due to increased computational usage of researchers working on this challenge.

## 7   Conclusion

We introduced *PulseImpute*, a novel imputation challenge for pulsative mHealth signals. We curated a set of ECG and PPG datasets with realistic mHealth missingness patterns and relevant downstream tasks. Our comprehensive set of baselines includes a novel Bottleneck Dilated Convolutional (BDC) transformer architecture that is able to exploit the quasiperiodicity present in our data and defines the SOTA. At the same time, our findings demonstrate that previous existing methods fail to achieve high performance, pointing out the need for additional research. PulseImpute addresses a significant gap in mHealth pulsative signal imputation, providing the first large-scale reproducible framework for the machine-learning community to engage in this unique challenge.

## Acknowledgements

We would like to thank Catherine Liu for her help and support on this work. This work is supported in part by NIH P41-EB028242-01A1, NIH 7-R01-MD010362-03, and the National Science Foundation Graduate Research Fellowship under Grant No. DGE-2039655. Any opinion, findings, and conclusions or recommendations expressed in this material are those of the authors and do not necessarily reflect the views of the National Science Foundation.

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
