# OpenReview forum: "PulseImpute: A Novel Benchmark Task for Pulsative Physiological Signal Imputation"
_NeurIPS.cc/2022/Track/Datasets_and_Benchmarks — NeurIPS 2022 Datasets and Benchmarks _

### Official Review · Reviewer_53jC · 2022-07-11
**A valid contribution to the sensory data imputation benchmark, together with a new imputation method**

**Rating:** 6
**Confidence:** 3

**Strengths:**

- The topic of signal sensor imputation is an important, realistic, and very practical problem in mHealth daily physiological data collection.
- Both the raw construction error and the three downstream tasks are valid experiment designs.
- The design of the new architecture does leverage the pattern of ECG/PPG properties and the advantage over the baseline methods are encouraging.

**Weaknesses:**

- Physiological signal types may go beyond ECG/PPG.
While ECG/PPG is arguably one of the most commonly collected physiological signals in mHealth applications. There are other common sensors that are not covered in this paper, such as IMU (accelerometer, gyroscope, and magnetometer), GSR, EEG, etc. I am not arguing that the authors must evaluate their techniques on these signals. But their characteristics could be very different from ECG/PPG signals. For example, they may not have a clear pulsative pattern. I am curious to know the authors' consideration of the generalizability of the technique and the potential necessity to tone down the paper's framing (or clarify the scope of the paper).

- Baseline method selection.
The authors compare the new technique against eight baseline techniques, which is great. However, why not compare against the two existing papers that specifically focus on imputing mHealth pulsative signals (i.e., [20, 40] - the citation number in the paper)? Please justify.

- Extremely low recall for baseline technique.
Related to the previous point, in Table 2, the performance of the baseline techniques all have very low recall (thus low F1 score). The authors provided some reasons in the text, which is great. But such a low performance raises a concern that whether these baselines are too easy to beat. Comparing against the SOTA technique could provide more valid results.

[20] Arman Iranfar, Adriana Arza, and David Atienza. Relearn: A robust machine learning framework in presence of missing data for multimodal stress detection from physiological signals. arXiv preprint arXiv:2104.14278, 2021.

[40] Hillol Sarker, Matthew Tyburski, Md Mahbubur Rahman, Karen Hovsepian, Moushumi Sharmin, David H Epstein, Kenzie L Preston, C Debra Furr-Holden, Adam Milam, Inbal Nahum-Shani, et al. Finding significant stress episodes in a discontinuous time series of rapidly varying mobile sensor data. In Proceedings of the 2016 CHI conference on human factors in computing systems, pages 4489–4501, 2016.

**Additional Feedback:**

Please see my comments above. I look forward to seeing the authors' response!

**Clarity:**

The paper is written well and easy to follow. The figures clearly convey the right message.

**Correctness:**

My concern of the pulsative signals only being a part of the wide range of physiological sensing signals has been mentioned above. Other than that, I do not have concern on the correctness of the paper. Evaluation methods and the experiment design are appropriate and valid.

**Documentation:**

The code has a clear structure and supports easy reproducibility.

**Ethics:**

The paper leverage publicly available datasets, thus I do not have ethical concerns of the paper.

**Relation To Prior Work:**

The paper covers a good amount of related work. Table 1 is very helpful to highlight the distinction of this work. My only concern is mentioned above in terms of selecting the right baseline techniques to compare.

**Summary And Contributions:**

In this paper, the authors proposed a new benchmark task for physiological sensor signal imputation. Specifically, the authors focus on ECG and PPG signals and use public datasets for evaluation. They first simulated realistic missing patterns for ECG and PPG signals by ablating samples and built a dataset with realistic missing data. They implemented eight existing traditional or modern time-series imputation techniques. Moreover, they also proposed and developed a new bottleneck dilated convolution (BDC) self-attention architecture that fits the characteristic ECG/PPG data: these data usually have a long-range structure as pulsative signals. The authors then evaluated the performance of these algorithms on 1. the raw signal reconstruction task, and 2. three downstream tasks: a) heartbeat detection in ECG, b) heartbeat detection in PPG, and c) cardiac pathophysiology classification in ECG. Their results indicate that the new BDC technique is significantly better than all baselines.

---

> ### Author Response · Authors · 2022-08-12
> **Response to Reviewer 53jC (1/1)**
>
> Thank you for the kind words and great discussion questions. We have renamed and reframed our paper around pulsative signals specifically, as well as clarified information about the baselines in the revised version of our paper. We address each of your questions in detail below. Please let us know if you have any further questions or need any additional clarifications.
>
> **Weaknesses**
>
> - **"Physiological signal types may go beyond ECG/PPG"** : We have modified the title to be "PulseImpute: A Novel Benchmark Task for Pulsative Physiological Signal Imputation" in our revised paper and further clarified within the paper to reflect that the scope of our work specifically focuses on "pulsative" signals. Pulsative signals are an exciting domain for imputation because they create a novel context for the development of ML methods in comparison to prior imputation tasks. There is rich information to be gained by modeling morphological structure over quasi-periods, and the accuracy with which signal morphology can be recovered has a direct impact on the performance of downstream tasks. Moreover, the fact that signal morphologies are well-defined (e.g. QRST complex in ECG) facilitates the interpretation of reconstruction results. We plan to add other pulsative signal types, such as Arterial Blood Pressure, to a future release of PulseImpute. Other physiological signals discussed (e.g. IMU, EEG, GSR) do not exhibit the pulsative property, and thus are not as well matched to  our area of focus.
> - **Why not Iranfar et al., 2021 and Sarker et al., 2016?**: Iranfar et al., 2021 and Sarker et al., 2016's primary goal was to develop an mHealth system framework for stress detection with multiple sensing modalities (i.e. ECG, PPG, respiration, electrodermal activity in Iranfar et al., 2021 and ECG, respiration, accelerometry in Sarker et al., 2016), and as such, they used multivariate imputation techniques (i.e. Multivariate Iterative Imputer and KNN, respectively). These multivariate methods are not applicable to our univariate setting, in which we seek to model information across time rather than across modalities.
> - **"Extremely low recall for baseline technique":** We have addressed this point by adding a new FFT-based baseline (Rahman et al., 2015), with the suggestion from Reviewer 7UND, giving new results shown in Figure 1, Table 2, Figure 3, and Table A3 in the revised manuscript. This method has the second highest Recall and  F1 score in the ECG Heartbeat Detection Task, achieving stronger performance than several deep learning based methods, as seen in Figure 1 and Table 2. We want to add that there is a general lack of strong baselines because pulsative imputation has not been studied extensively prior to our work. A few additional comments:
>   - Many prior mHealth works were limited to simple imputation methods as their focus was not primarily on imputation (e.g.  Iranfar et al., 2021 and Sarker et al., 2016, see discussion above).
>   - We have included multiple SOTA benchmarks from the general time-series imputation literature: BRITS (SOTA w/ public code in PhysioNet 2012, UCI localization), DeepMVI (SOTA w/ public code in 10 time-series datasets, spanning from sales to weather data), NAOMI (SOTA w/ public code in Billiards Ball Trajectory, Basketball Tracking, PEMS-SF Traffic). These methods did not perform well in our setting, presumably because they were not designed to exploit the unique properties of pulsative signals. This motivated our development of the additional BDC Transformer baseline.
>
> **References**
>
> Iranfar, A., Arza, A., & Atienza, D. (2021, November). Relearn: A robust machine learning framework in presence of missing data for multimodal stress detection from physiological signals. In 2021 43rd Annual International Conference of the IEEE Engineering in Medicine & Biology Society (EMBC) (pp. 535-541). IEEE.
>
> Sarker, H., Tyburski, M., Rahman, M. M., Hovsepian, K., Sharmin, M., Epstein, D. H., ... & Kumar, S. (2016, May). Finding significant stress episodes in a discontinuous time series of rapidly varying mobile sensor data. In _Proceedings of the 2016 CHI conference on human factors in computing systems_ (pp. 4489-4501).
>
> Rahman, S. A., Huang, Y., Claassen, J., Heintzman, N., & Kleinberg, S. (2015). Combining Fourier and lagged k-nearest neighbor imputation for biomedical time series data. Journal of biomedical informatics, 58, 198-207.

---

### Official Review · Reviewer_nyqJ · 2022-07-20
**Review – A solid challenge and benchmark for a relevant problem**

**Rating:** 8
**Confidence:** 3

**Strengths:**

The proposed datasets and challenges appear highly relevant to facilitate broad usage of wearable sensors for medical purposes. The definition of evidenced-based patterns of missingness fills a gap in previously published work to that end.
That definition is made accessible by the proposed challenge, which is based on existing (and peer-reviewed) datasets, incorporating the patterns missingness. The experiment design is clear and thoughtful. The downstream taks are both difficult and relevant tasks for pulsative signals to test imputation methods. The data provided makes it easy for researchers to work on the topic. The authors propose a model architecture which demonstrates significant outperformance on the benchmark tasks.

**Weaknesses:**

In my view, there are no major weaknesses. Two minor issues:

First, the presentation of the BDC architecture may be a bit too confident. In ll. 211 following the authors state their requirements for their benchmark model and state that ‘no existing transformer models address all three issues sufficiently’. I’d argue that these are rather common issues. As an example: images / pixels require local context, require some measure against permutation equivariance and have to deal with scaling of long sequences. [Vision transformers](https://arxiv.org/abs/2010.11929) (ViT) have found ways to deal with that, so has the [Perceiver](https://arxiv.org/abs/2103.03206) architecture, directly on the data. [CvT](https://openaccess.thecvf.com/content/ICCV2021/papers/Wu_CvT_Introducing_Convolutions_to_Vision_Transformers_ICCV_2021_paper.pdf) adds convolution data encoders to ViT, similar to the author’s approach. That does not take anything away from their model architecture, which they show works fine. I’d find it fair though to put it into context of other work. The authos further state that positional encodings don’t have a good inductive bias in their setting (line 239). I’m curious, is that empirical or are there other reasons to think that? On other domains (again, images as example), position encodings work surprisingly well to contextualize.

Second, equation 1) seems somewhat over-complicated. Self-attention is commonplace and a softmax would have simplified it greatly.

**Additional Feedback:**

None

**Clarity:**

The paper is very well written. The flow from motivation over the background to the challenges and benchmark make it a rewarding read from which I could gain something.
The only minor issue:  I had a few questions on the exact ‘realistic patterns of missingness’ (l 39) until the authors explained where they come from in Section 3. Maybe that can be caught a little earlier?


**Correctness:**

Dataset, benchmark tasks as well as paper seem of very good quality. The motivation is made clear, design decisions for the dataset selection, creation of missingness or benchmark architecture are explained and justified. The benchmark results seem sound and convincing.
If there is space, I would appreciate more insights or discussion of the results on extended missingness (Figure 3, right). Based on the plots, BDC appears much richer in information than, e.g., BRITS, which nonetheless seems to work well on the Diagnostic label group. Are those spurious correlations, or features carried over by BRITS from the true signal? Vice-versa, do the authors have insights into what mode of information is lost by BDC that could explain the particular drop in predictive performance?


**Documentation:**

The datasets used for the challenge have been published in previous work. Data, challenge and pre-trained models are readily available on a public github and well documented.


**Ethics:**

The orginal datasets used for the challenge appears to contain both medical information of patients as well as geo-location, which may be critical. However, the datasets have already been cleared by ethics comittees and been published. Re-using the data does not introduce new ethical concerns to me.

The datasets included in this submission contain only the pulsative signals like ECG and PPG and thus on their own don't pose ethical concerns.

**Relation To Prior Work:**

The proposed dataset and task appear novel and original. Previous work on health data imputation either worked on non-pulsative signals, was not publicly available, did not use realistic missingness patterns or could not be used as benchmarks.

**Summary And Contributions:**

The usage of wearable sensors for medical purposes promises better monitoring with high frequency information. However, the usage of such sensors worn in day-to-day life often leads to gaps in the sensory information. Imputation techniques attempt to fill such gaps, but are lacking for pulsative signals like ECGs and PPGs. This submission supplies datasets, methods to mimic realistic ‘missingness’ in the data, as well as challenge tasks to evaluate pulsative signal imputation. Further, the authors propose a benchmark transformer model, where the signal tokenizer is realized via dilated convolution, and empirically demonstrate significant outperformance compared to previous work on their benchmark tasks.

---

> ### Author Response · Authors · 2022-08-12
> **Response to Reviewer nyqJ (1/1)**
>
> Thank you for the kind words and great discussion questions. We have modified our framing of BDC as suggested, as well as added our discussions into the revised paper. We address each of your questions in detail below. Please let us know if you have any further questions or need any additional clarifications.
>
> **Weaknesses**
>
> - **"presentation of the BDC architecture may be too over-confident"** : We have toned down the wording in the revised paper.
> - **"[why] positional encodings don't have a good inductive bias"**: We have a few reasons to believe this, which we have included in the Appendix A4, and we briefly describe below:
>   - Intuitively, the absolute position relative to the start of a pulsative signal is not meaningful, due to arbitrary time at which sensors are attached  and begin recording. Specifically, recording could start at any time relative to a given heartbeat.
>   - Even if the signals were aligned at the start, due to phase variance from the heart rate variance phenomena, the relative position of specific waveform shapes will vary.
>   - In contrast to the above, images used in computer vision are often framed so that the subject of the photograph tends to be located in the center of the image (e.g. the widely-used ImageNet dataset has this property). In this setting, positional encoding is likely to be more useful as the background context would tend to occupy specific image areas consistently.
>   - We also note that empirically, we can see that the introduction of positional embedding degrades performance.
> - **"equation 1) seems somewhat over-complicated"**: We have simplified the equation in the revised paper.
>
> **Correctness**
>
> - **"discussion of the results on extended missingness":** This is a great discussion point, and we have added it in lines 308-314. We believe that these results may be tied to the BDC transformer reconstructing much shorter R peaks in the ECG signal under high missingness percentages, which can be more clearly seen in Figure A14 found in the Appendix. Some of the diagnostic labels are dependent on R peak height (e.g. Left Anterior Fascicular Block is diagnosed with tall R waves in Lead I (Larkin & Buttner, 2021)), and thus will be adversely affected. We hypothesize that the minimalistic imputations produced by BRITS, Conv9 and mean methods (see Figure 3 and A14) fare better because there is less misleading and ambiguous signal information present. We leave further analysis to future work, with techniques such as attribution maps as a potential avenue to explore (as done in Fig 8 of (Strodthoff et al., 2020)).
>
> **References**
>
> Larkin, J. &  Buttner, R. (2021, February 7). Left anterior fascicular block (LAFB). Life in the Fast Lane • LITFL. Retrieved August 7, 2022, from https://litfl.com/left-anterior-fascicular-block-lafb-ecg-library/
>
> Strodthoff, N., Wagner, P., Schaeffter, T., & Samek, W. (2020). Deep learning for ECG analysis: Benchmarks and insights from PTB-XL. IEEE Journal of Biomedical and Health Informatics, 25(5), 1519-1528.

---

> > ### Comment · Reviewer_nyqJ · 2022-08-16
> > **Thanks for the detailed responses and revision.**
> >
> > I would like to thank the authors for the response. I appreciate that the authors considered the feedback of five reviews in their revision of the paper, which appears to have been a great deal of work. The minor weaknesses and questions I had are adequately addressed, which improves the readability of the paper and confirms me in my score.
> >
> > Further, I appreciate the thoughts regarding position encodings. It’s really just a minor thing that can be addressed in future revisions, but I stumbled over lines 449-452 in the revised manuscript.
> > Local context and scaling matters not only in time series data, but also in images or in natural language, where the order of words can define the semantic concept. I.e., “The table is on the box” vs. “The box is on the table.” Also, not all semantic concepts are centered. In image segmentation, not just the object in the center matters, but local context on the entire image. On these domains and tasks, position encodings have been applied not only to break permutation equivariance, but also to encode distance relations and thus local context. The commonly used sine-like signals at different frequencies explicitly aim at resolving the different relative positions.
> > To me, applying either position encodings or using convolutions can be used to include local context. In this particular setting, convolutions seem to offer a larger receptive field / higher parameter efficiency, which is a good reason to use them. Adding position encodings to BDC just disturbs the signal, reduced performance with position encodings is therefore plausible. Maybe using convolutions over position encodings can be framed as a design decision?

---

> > > ### Author Response · Authors · 2022-08-19
> > > **Follow-up Response to Reviewer nyqJ**
> > >
> > > Thank you so much for your kind words. This is a great point, and we are in agreement. In the final version of the paper, we will reframe our discussion of not using positional encoding as a design decision.

---

### Official Review · Reviewer_Uavi · 2022-07-21
**Interesting topic, but needs better presentation**

**Rating:** 6
**Confidence:** 4

**Strengths:**

- This paper defines an interesting research question.
- Code to replicate the baseline results.

**Weaknesses:**

- I can see why the authors chose to apply the missingness patterns in the mHealth settings to the existing pulsative signal datasets, rather than collecting data directly from mHealth wearables (due to a possible lack of ground truth). Admittedly, the authors also compared several differences between data collected from mHealth settings and clinical settings. However, the lack of a direct comparison of the differences in missing patterns in the data collected from the two settings, and what exactly the missing patterns are in the data collected from mHealth wearables, makes the choice of curated datasets to address the question raised in this paper less convincing.
- What makes pulsative signals more interesting/compelling than other signals from a missingness perspective was less motivated.
- The paper is a bit hard to follow and there are some inconsistencies and inaccessible references.


**Additional Feedback:**

Following are several minor issues to be addressed.
- In section 2 (related work), instead of reviewing prior work and highlighting its limitations simultaneously in each subsection, the authors may consider systematically reviewing prior work in each subsection, and then summarizing the issues with existing work in the last paragraph of this section.
- In section 2 (related work) line 61, EHR needs to be specified.
In section 2 (related work) line 75, LOCF was mentioned, but I’m confused about which work uses LOCF for imputation. Also, LOCF needs to be specified (i.e., Last Observation Carried Forward).
- In section 3.1 line 128, ABP needs to be specified.
Some references are hard to access. For example, [24] in line 377 and [33] in line 397, I’m not sure which paper/work the authors are referencing.
- In section 3.1 lines 145-146, “the majority of” and “some of” should be quantified in this sentence “the majority of missingness gaps are 3-9 seconds long but some gaps can last more than a minute.”. Specifically, whether the majority is 90% or 70%?
- Some statements in this paper need references. For example, in section 3.1 lines 134-135, “We sampled 5 min waveforms, as this is a standard duration for tasks like HRV that are based on beat detection”. As another example, in section 4 lines 208-210, this sentence needs a reference “Our Our primary focus is on transformer-based approaches because self-attention is an attractive method for modeling the long-range structure of pulsative signals as a means to leverage quasi-periodicity in imputing large blocks of missing samples.”.


**Clarity:**

This paper is a bit hard to follow. Specifically,
- it reads to me that PulseImpute is a combination of a set of datasets for imputation tasks, nine baseline imputation models (including eight replicated existing models + one newly proposed model), and downstream task performance benchmarks. So in the last paragraph of section 1, it might be better to state that the main contribution is PulseImpute, and then to detail the three sub-contributions. In addition, I’m a bit confused as to whether the nine baseline models mentioned in line 51 contain the one proposed by the authors. Referring to Table 2, it does seem to contain the newly proposed imputation method. Then, I suggest reversing the second and third bullets in lines 51 and 52 and making it clear that the nine baseline models include the eight existing imputation methods and your new proposed method.
- there are some inconsistencies in this paper. For example, in sections 3.1, 3.2, and 3.3, the authors said that F1 score is used to measure classification performance. However, in Table 2, they reported F1, precision, and sensitivity, and in Figure 3, they reported AUC. As another example, the authors used nine models as baselines, however, in figure 1, it only contains visualization of imputation results from six baselines without explaining why the other three baselines were left out (although the authors do show the visualization of imputation results from all baselines in the appendix).
- some of the statements in the paper should be more precise. For example, in lines 38-39, the authors claim that “Table 1 describes six criteria that the PulseImpute challenge provides which are lacking in prior work.”. In fact, all six criteria were partially presented in prior work, but no work covered these six criteria simultaneously.
- some parts might need to be reorganized. For example, In section 3 (PulseImpute Challenge Description), lines 101-111 read to me like a review of two datasets the authors used for further curation, should it be part of related work? In addition, lines 112-119 look like brief summaries of each section of the paper. It seems better to put them in the introduction.


**Correctness:**

I have a few concerns about the correctness of the methods used in this paper.
- Although the authors discussed their motivation for using clinical pulsative waveform datasets to mimic mHealth pulsative waveforms in A1, I’m still somewhat concerned as there is no direct comparison in missing patterns between pulsative signals collected from wearables and clinical settings to support their choice is an appropriate approach.
- As the paper applies missingness patterns from real-world mHealth settings to the existing pulsative signal datasets, it is crucial to have a description/review of missingness patterns the authors used to help reviewers/readers to justify this application is reliable. However, I referred to references [7] and [29] and supplemental materials but did not find any description of the missingness patterns.
- I wonder if the data for the third downstream task (predict diagnosis) is balanced? If not, they may want to add balanced accuracy and kappa as metrics to evaluate performance.


**Documentation:**

All data is available, but descriptions are lacking. The Github repo provided in this paper is sufficient to support reproducibility.


**Ethics:**

There are no ethical concerns to raise.

**Relation To Prior Work:**

This paper identified a gap in the existing literature and made efforts to fill it.


**Summary And Contributions:**

This paper raises the issue of missing data in pulsative signals collected from wearable devices and introduces an imputation benchmark (PulseImpute). Specifically, the authors extracted missingness patterns from real-world mHealth settings and applied them to two existing pulsative signal datasets. They reproduced several existing imputation methods and proposed a new imputation method and used them as baselines. Applying these baseline imputation methods to their processed pulsative signal datasets with missing values, the authors proposed benchmarks for the downstream tasks.

---

> ### Author Response · Authors · 2022-08-12
> **Response to Reviewer Uavi (1/3)**
>
> Thank you for the detailed review and great discussion points. We have gone through the paper in detail again for reassessment based upon the points you have raised and added our discussions into the revised paper. We address each of your questions in detail below. Please let us know if you have any further questions or need any additional clarifications.
>
> **Weakness + Correctnesss**
>
> - **Concern about using clinical signals and mHealth missingness:** As noted, our approach of applying mHealth missingness patterns on clean clinical pulsative signals is necessary for two reasons: 1) Due to the lack of signal ground truth to evaluate imputations directly on mHealth datasets; and 2) Because mHealth datasets are too small to be used in a comprehensive benchmark for deep learning models (e.g. PPG-DaLiA, an mHealth dataset, has 15 subjects,  whereas our curated MIMIC-III PPG dataset, a clinical hospital dataset, has 18,210 subjects). Previous work has demonstrated that mHealth ECG sensors are able to record clinically-accurate single lead ECG tracings in both healthy subjects and subjects with underlying cardiac disease or rhythm abnormalities (Kathleen et al., 2017; Gropler et al., 2018) and  rhythm-based and morphological-based metrics can be calculated from PPG sensors, regardless of setting (Nardelli et al., 2020; Rajala et al., 2018). These findings suggest that domain gap issues between clinical and mHealth settings, while they exist, are not a major obstacle.
> - **Further clarification on missingness:** Below we address additional concerns raised by the reviewer:
>   - "**Lack of [discussion on] differences in missing patterns in the data collected from [mHealth and clinical]"**: Our focus is on addressing mHealth missingness using clinical data, and we explicitly curate it so that there is no missingness, prior to systematically applying our missingness models in our experiments. The curation procedure to remove signals with missingness for the MIMIC-III datasets is described in detail in Appendix A2. For the PTB-XL dataset, there was a quality assessment by a technical expert in its original release (Wagner et al., 2020), and further missingness-removal curation was unnecessary. The analysis of missingness within clinical waveform data can be a topic for future research.
>   - **"What … the missing patterns are in … mHealth wearables" and "description/review of missingness patterns …** **I referred to references of  (Chatterjee et al., 2020, Reiss et al. 2019 )"**: We are using 4 types of missingness patterns in our paper: extracted ECG mHealth missingness, extracted PPG mHealth missingness, Extended Loss, and Transient Loss. We briefly review them below, and have clarified their descriptions in the revised text, in Sections 3.1, 3.2, and 3.3.
>     - The extracted ECG/PPG mHealth missingness are the real missingness patterns  that occurred during the two mHealth studies (Chatterjee et al., 2020, Reiss et al. 2019) respectively. Because the focus of these two studies was on developing mHealth biomarkers, their papers do not describe the missingness present in their data. In Appendix A3, we include multiple visualizations of the ECG/PPG mHealth missingness patterns, including a histogram of the distribution of missingness gaps lengths. We have included the extracted missingness patterns in our data release, and this will allow for users to visualize and explore the patterns and support future work on modeling missingness, for example as a stochastic process.
>     - The Extended and Transient Loss patterns are used to simulate common paradigms found in mHealth. We describe the generative procedure in Section 3.3 and use the work done by Rahman et al., 2017, one of few works to systematically study mHealth missingness, as the basis for our approach. Specifically, one can refer to Fig 1 in Rahman et al., 2017 for the major sources of data loss.

---

> > ### Author Response · Authors · 2022-08-12
> > **Response to Reviewer Uavi (2/3)**
> >
> > **Weakness + Correctnesss (continued)**
> > - **"What makes pulsative signals more interesting/compelling than other signals from a missingness perspective was less motivated":** This is a great discussion point, and we included it in Lines 32-37. We believe that the quasi-periodic structure of pulsative signals creates a novel context for the development of ML methods in comparison to prior imputation tasks. Specifically, there is rich information to be gained by modeling morphological structure over time. Additionally, the accuracy with which signal morphology can be recovered has a direct impact on the performance of downstream tasks. The fact that signal morphologies are well-defined (e.g. the QRST complex in ECG) facilitates the interpretation of reconstruction results. Finally, while essentially all imputation works use simulated missingness to obtain ground truth accuracy, in our mHealth context we can leverage both extracted real-world missingness patterns (see lines 153-157, 175-178) and domain specific simulated missingness (e.g. transient loss to mimic packet loss, see lines 196-210), thereby increasing both the complexity and realism of our simulated missingness experiments.
> >
> > - **Performance Metrics for Imbalanced Classes in Diagnosis Cardiac Classification:** To clarify, this is a multi-label classification problem (i.e., the label space is not disjoint). This leads to an imbalanced distribution in the Diagnostic category, which motivates our use of the Macro-AUC to quantify generalization performance. Macro-AUC is a common measure for the multi-label classification problem under label imbalance (Spyromitros-Xioufis, 2011;  Daniels & Metaxas, 2017; Jamthikar et al., 2022) and has been theoretically proven to be optimized when the instance-wise margin is maximized (Wu & Zhou, 2017). The larger the margin, the more effectively the classifier is able to distinguish between relevant and irrelevant labels for a given instance. We have revised the paper to include this discussion (see lines 216-219)
> > - **Data Descriptions are lacking:** Fixed, we have added more information within the README.md, which can be found in the data dropbox's root folder.
> >
> > **Clarity + Minor issues** :
> >
> > - **Organization of last paragraph of section 1:** Fixed and updated number of baselines to reflect new FFT-based imputation baseline.
> > - **F1 Score versus AUC:** To clarify, the F1 Scores are metrics for the heartbeat detection task and the Macro-AUC Scores are the metrics for the multi-label cardiac classification task. We use F1 score for the heartbeat detection task because it is commonly used in peak detection problems (Liu et al., 2018; Cai et al., 2020, Malik et al., 2020) in order to summarize information about peaks correctly/incorrectly detected and undetected. We justify the usage of Macro-AUC for the cardiac classification task above in the "Performance Metrics for Imbalanced Classes in Diagnosis Cardiac Classification" section. We have further clarified these metrics in our revised text in lines 162-164, 216-219.
> > - **Subset of Results in Figure 1:** Fixed to include all results with newly increased page limit.
> > - **Six criteria in Table 1:** Fixed
> > - **Unnamed abbreviations and Clarity:** Fixed
> > - **Broken References:** Fixed
> > - **Clarity in mHealth Systems' Imputation:** Fixed
> > - **Quantifying Missingness Gaps:** Fixed
> > - **Need References:** Fixed the first suggestion, but "self-attention is an attractive method for modeling the long-range structure of pulsative signals as a means to leverage quasi-periodicity in imputing large blocks of missing samples" is not prior work: this is a claim that we make and justify throughout Section 4 and in Appendix A4 (Experiments Validating BDC Self-Attention Design). We clarify this statement in the revision in lines 239-241.

---

> > > ### Author Response · Authors · 2022-08-12
> > > **Response to Reviewer Uavi (3/3)**
> > >
> > >
> > > **References**
> > >
> > > Kathleen, T. H., Angelo, B. B., Garan, H., Sciacca, R. R., Riga, T., Warren, K., ... & Whang, W. (2017). Evaluating the utility of mHealth ECG heart monitoring for the detection and management of atrial fibrillation in clinical practice. _Journal of atrial fibrillation_, _9_(5).
> > >
> > > Gropler, M. R., Dalal, A. S., Van Hare, G. F., & Silva, J. N. A. (2018). Can smartphone wireless ECGs be used to accurately assess ECG intervals in pediatrics? A comparison of mobile health monitoring to standard 12-lead ECG. _PLoS One_, _13_(9), e0204403.
> > >
> > > Nardelli, M., Vanello, N., Galperti, G., Greco, A., & Scilingo, E. P. (2020). Assessing the quality of heart rate variability estimated from wrist and finger ppg: A novel approach based on cross-mapping method. Sensors, 20(11), 3156.
> > >
> > > Rajala, S., Lindholm, H., & Taipalus, T. (2018). Comparison of photoplethysmogram measured from wrist and finger and the effect of measurement location on pulse arrival time. _Physiological measurement_, _39_(7), 075010.
> > >
> > > Wagner, P., Strodthoff, N., Bousseljot, R. D., Kreiseler, D., Lunze, F. I., Samek, W., & Schaeffter, T. (2020). PTB-XL, a large publicly available electrocardiography dataset. _Scientific data_, _7_(1), 1-15.
> > >
> > > Chatterjee, S., Moreno, A., Lizotte, S. L., Akther, S., Ertin, E., Fagundes, C. P., ... & Kumar, S. (2020). Smokingopp: Detecting the smoking 'opportunity' context using mobile sensors. Proceedings of the ACM on interactive, mobile, wearable and ubiquitous technologies, 4(1), 1-26.
> > >
> > > Reiss, A., Indlekofer, I., Schmidt, P., & Van Laerhoven, K. (2019). Deep PPG: Large-scale heart rate estimation with convolutional neural networks. Sensors, 19(14), 3079.
> > >
> > > Rahman, M., Ali, N., Bari, R., Saleheen, N., al'Absi, M., Ertin, E., ... & Kumar, S. (2017). mdebugger: Assessing and diagnosing the fidelity and yield of mobile sensor data. In Mobile Health (pp. 121-143). Springer, Cham.
> > >
> > > Spyromitros-Xioufis, E., Spiliopoulou, M., Tsoumakas, G., & Vlahavas, I. (2011, June). Dealing with concept drift and class imbalance in multi-label stream classification. In _Twenty-Second International Joint Conference on Artificial Intelligence_.
> > >
> > > Daniels, Z. A., & Metaxas, D. N. (2017, February). Addressing imbalance in multi-label classification using structured hellinger forests. In _Thirty-First AAAI Conference on Artificial Intelligence_.
> > >
> > > Jamthikar, A., Gupta, D., Johri, A. M., Mantella, L. E., Saba, L., & Suri, J. S. (2022). A machine learning framework for risk prediction of multi-label cardiovascular events based on focused carotid plaque B-Mode ultrasound: A Canadian study. Computers in Biology and Medicine, 140, 105102.
> > >
> > > Wu, X. Z., & Zhou, Z. H. (2017, July). A unified view of multi-label performance measures. In _international conference on machine learning_ (pp. 3780-3788). PMLR.
> > >
> > > Liu, F., Liu, C., Jiang, X., Zhang, Z., Zhang, Y., Li, J., & Wei, S. (2018). Performance analysis of ten common QRS detectors on different ECG application cases. _Journal of healthcare engineering_, _2018_.
> > >
> > > Cai, W., & Hu, D. (2020). QRS complex detection using novel deep learning neural networks. IEEE Access, 8, 97082-97089.
> > >
> > > Malik, J., Soliman, E. Z., & Wu, H. T. (2020). An adaptive QRS detection algorithm for ultra-long-term ECG recordings. _Journal of Electrocardiology_, _60_, 165-171.

---

> > > > ### Comment · Reviewer_Uavi · 2022-08-18
> > > > **Thank you for the detailed responses and revision, I still have one more concern**
> > > >
> > > > Thanks to the authors for their efforts in this revision. Most of my concerns were well addressed. However, there is still one more concern remaining. In this paper, the authors apply the mHealth missingness patterns to the existing pulsative signals collected from clinical settings. I think it’s important to demonstrate that the missingness patterns of data collected in mHealth and clinical settings are similar. For example, missingness occurs when the sensors fail to sense the physical environment. For data collected in the mHealth settings, missingness patterns can be multiple missing segments that are sometimes not sensed by the sensor due to physical activities over a period of time (e.g., running). For data collected in the clinical settings, since patients are relatively stationary, the missingness pattern may be a whole period of missing caused by the sensor being far away from the body and unnoticed. The authors may need to demonstrate that this is not the case.

---

> > > > > ### Author Response · Authors · 2022-08-19
> > > > > **Follow-up Response to Reviewer Uavi**
> > > > >
> > > > > Thank you for your kind words. Regarding your remaining question, there are similarities in missingness patterns across the mHealth and clinical domains. For example, with respect to  participant compliance, both clinical patients and mHealth users can remove sensors, resulting in contiguous blocks of missing data. Likewise, participant movement in both contexts can result in artifacts (e.g. tugging at a finger-mounted sensor in the hospital, vs adjusting the strap of a mHealth wearable which becomes uncomfortable). At the same time, the mHealth environment is more challenging for data capture and may experience more missingness overall.
> > > > >
> > > > > Also, we just wanted to clarify that the comparisons between clinical and mHealth missingness do not affect our findings and experimental design because:
> > > > > * We do not make any claims about the suitability of our approach for addressing the issue of clinical missingness.
> > > > > * There is no clinical missingness present in our benchmark dataset
> > > > >
> > > > > Our contribution is on introducing a benchmarking suite for pulsative signals with realistic mHealth missingness, and our data curation process (described in Sections 3.1, 3.2, Appendix A2, and Response 1/3) ensured that signals with clinical missingness (e.g. the sensor being away from the body and unnoticed, as raised by the reviewer) were removed from the dataset.
> > > > >
> > > > > Based on the reviewer’s suggestion we will add an additional characterization of clinical missingness patterns to the final paper.

---

### Official Review · Reviewer_GtXW · 2022-07-27
**PulseImpute: A Novel Benchmark Task for Physiological Signal Imputation**

**Rating:** 6
**Confidence:** 3

**Strengths:**

The authors demonstrate that the current state-of-the-art algorithms trained on ECG and PPG waveform perform poorly on signals obtained from wearables.
Table 1 outlines a clear comparison to existing work. The explanation why multi-channel is out of scope is solid.


**Weaknesses:**

The approach was only tested on one dataset for each domain. Validation on other datasets would strengthen the paper.

**Additional Feedback:**

The submission is better suited for the main conference.

**Clarity:**

The paper is clearly written but not quite appropriate for the dataset and benchmark track.

**Correctness:**

The authors' claim to have created the largest publicly-available ECG and PPG waveform dataset is inaccurate.  The project is more of a transformation of existing datasets rather than new data.
The use of KNN or other non-deep learning techniques is not a disadvantage per se as the authors seem to suggest. Specific algorithmic limitations that cannot be solved with KNN or similar methods should be elaborated further. A deep learning approach is not necessarily better in all KPIs.

**Documentation:**

The authors published their model on GitHub, a win for reproducibility.

**Relation To Prior Work:**

The submission failed to discuss how the approach presented compares to previous work when it comes to explainability of predictions.

**Summary And Contributions:**

An existing architecture was slightly transformed and transferred to another domain (i.e. wearables) to address a gap in physiologic signal imputation.  The approach presented shows superior performance compared to existing work.

---

> ### Author Response · Authors · 2022-08-12
> **Response to Reviewer GtXW (1/2)**
>
> Thank you for your great discussion points. We have addressed each of your points below and added this discussion into our revised paper. Please let us know if you have any further questions or need any additional clarifications.
>
> - **"The authors demonstrate that the current state-of-the-art algorithms trained on ECG and PPG waveform perform poorly on signals obtained from wearables."**: We just want to clarify that all of our experiments are done on signals collected under clinical conditions, not on signals from wearables. It is the missingness patterns that are taken from real-world wearable data, enabling us to construct a dataset with real-world missingness patterns and with ground truth so that imputation accuracy can be computed (see Section 3 for details).
> - **"The authors' claim to have created the largest publicly-available ECG and PPG waveform dataset is inaccurate. The project is more of a transformation of existing datasets rather than new data."**:
>   - First, we would like to clarify the wording in this claim. We have been careful to state that our claim is with respect to "curating" rather than "creating" our dataset (see lines 131, 168), to accurately reflect the nature of our contribution.
>   - Second, we believe that this comment understates the effort that was required to take these existing datasets and construct the first comprehensive dataset for pulsative imputation.  The MIMIC-III Waveform dataset is not easily usable due to its high noise content and large size (e.g. 6.7 TB and 5 mil single-lead ECGs).  All prior works using this dataset have not processed the waveforms in their entirety, choosing instead to process a small random subset (i.e. 1,000 2-minute ECG signals from 50 patients in Bashar et al., 2019) or to utilize the smaller subset with matching clinical data (i.e. 30,124 5-second ECG signals from 15,062 patients in Kamaleswaran et al., 2020), and these works do not make their curated data available online and lack code for reproducibility. In comparison, after performing the first complete curation of the MIMIC-III waveforms, we obtained 440,953 5-minute ECG waveforms from 32,930 patients. _This is orders of magnitude more usable data than any prior work_. This curation process required a significant amount of physiological signal domain knowledge and processing resources, and was a time-consuming effort by our team that we believe can benefit the research community significantly. We include details about our clean signal filtration process in Appendix A2 and released our preprocessing code in our Github repo.
> - **"Not appropriate for dataset and benchmark track"**: We don't believe this track is intended to be narrowly-defined around works that collect new data. Rather, the intention of this track is to provide a venue for "highly valuable machine learning datasets and benchmarks, as well as a forum for discussions on how to improve dataset development"  (NeurIPS Foundation, 2022), as opposed to the focus on methodological advancements found in the main conference.  The 2022 Call for Papers, states that "benchmarks on new or existing datasets'' are within scope (NeurIPS Foundation, 2022), and therefore, we believe that the creation of our novel benchmark task meets the criteria for inclusion. Please see Table 1, Introduction, and Related Work for a detailed discussion about how our work is the first to comprehensively provide a strong benchmarking suite for mHealth pulsative signal imputation.
> - **"The approach was only tested on one dataset for each domain":** We want to point out that in the case of the ECG domain we are actually using two different datasets, one curated from MIMIC-III Waveform and one from PTB-XL (see Sections 3.1 and 3.3).
> - **"The use of KNN or other non-deep learning techniques is not a disadvantage" and "failed to discuss how the approach presented compares to previous work when it comes to explainability of predictions."**: We agree that the fact that a method that does not employ deep learning is not necessarily at a disadvantage. The issue of explainability of deep models is a great discussion point, and we have added it to our Limitations section (see lines 336-338). Moreover, in response to this comment and a suggestion from Reviewer 7UND, we have added an FFT-based imputation method as an additional non-deep learning benchmark in our revised paper. This method can exploit the quasi-periodicity present in our pulsative signals and has stronger performance than several prior deep-learning methods in the Heartbeat Detection Tasks, which can be seen in Figure 1 and Table 2.

---

> > ### Author Response · Authors · 2022-08-12
> > **Response to Reviewer GtXW (2/2)**
> >
> > **References**
> >
> > Bashar, S. K., Ding, E., Walkey, A. J., McManus, D. D., & Chon, K. H. (2019). Noise detection in electrocardiogram signals for intensive care unit patients. IEEE Access, 7, 88357-88368.
> >
> > Kamaleswaran, R., Lian, J., Lin, D. L., Molakapuri, H., Nunna, S., Shah, P., ... & Padman, R. (2020). Predicting volume responsiveness among sepsis patients using clinical data and continuous physiological waveforms. In AMIA Annual Symposium Proceedings (Vol. 2020, p. 619). American Medical Informatics Association.
> >
> > NeurIPS Foudation. (2022). Call for NeurIPS 2022 Datasets and Benchmarks Track Papers. NeurIPS 2022 Datasets and Benchmarks Track. Retrieved August 11, 2022, from https://neurips.cc/Conferences/2022/CallForDatasetsBenchmarks#:~:text=NeurIPS%202022%20Datasets%20and%20Benchmarks,how%20to%20improve%20dataset%20development.

---

### Official Review · Reviewer_7UND · 2022-07-28
**Proposed dataset for benchmarking waveform imputation methods fills an important gap in the waveform imputation community.**

**Rating:** 7
**Confidence:** 4
**Clarity:** The paper is nicely written.

**Strengths:**

Strengths of the dataset include the scale and the realistic masking procedure used to generate missingness that tries to closely mirror patterns seen in mHealth systems. I also appreciate that a downstream prediction task is also included in the dataset, since often users care about how imputation quality will affect application performance. Many of the potential concerns I had with the dataset selection (e.g., using ICU patients, mHealth-specific missingness patterns, differences in ECG sensor quality from hospital to mHealth) were preemptively by the authors’ explanation in Appendix A1.

**Weaknesses:**

I’m not sure that mean and linear imputation necessarily make sense as baselines. If a peak is defined as a local maximum (i.e., $(x[i] > x[i-1])$ && $(x[i] > x[i+1])$ at index $i$ in signal $x$), then by definition there will be no peaks with mean or linear interpolation and the peak classification will fail. This may explain the NaN values and zero sensitivity of the mean and linear imputation in Table 2. Why not use a simple FFT-based baseline since these are quasi-periodic signals?

The choice to not utilize the MIMIC matched waveform database so that covariates can be used either in the modeling process or during evaluation is also a weakness. Many parameters such as resting heart rate, max heart rate, etc. vary as a function of age, disease status, medication usage, etc. and as a benchmarking dataset it would be nice to be able to incorporate that data.


**Additional Feedback:**

Questions

- Why did the authors not include confidence intervals in Table 2? In general, without confidence intervals it’s difficult to tell if proposed methods improve over baselines or state-of-the-art if only point estimates of performance metrics are given.

Comments
- In the Appendix (lines 28-29) the authors write “Because these paradigms are rhythm-based, the morphological differences are not relevant, only the timing of the signals, which is exactly what our PPG Heart Beat Detection task tests.” I disagree with this point - I think that if the goal is to measure quality of imputation, the reconstruction of morphology is important. If waveform morphology was irrelevant, then imputation algorithms should focus on estimating the average heart rate or beat-to-beat intervals (i.e., R-R interval), and likely downstream tasks such as atrial fibrillation classification wouldn’t benefit from using the raw waveform compared to derived features. I would advocate that the waveform imputation community consider adding morphology features such as T-wave identification in the ECG waveforms or left ventricle ejection time (LVET) in the PPG waveforms as target quantities of interest.

Suggestions
- One suggestion is for the authors add the number of patients in the datasets described in Sections 3.2 and 3.3.
- Another suggestion would be to break the training dataset into chunks in case users want to download a subset of the data without downloading such a huge file.
- An extension/addition the authors could consider is adding additional waveform data for a subset of users/patients such that models could be fine-tuned or personalized during an initial data gathering stage before being fully deployed (e.g., see Deng et al. 2021).


Deng, Y., Lu, L., Aponte, L. et al. Deep transfer learning and data augmentation improve glucose levels prediction in type 2 diabetes patients. npj Digit. Med. 4, 109 (2021). https://doi.org/10.1038/s41746-021-00480-x


**Correctness:**

What do the authors mean by “pooled all channels together” (line 131)? Are multiple types of ECG leads (e.g., I, II, III, aVR, V, etc.) included in the ECG dataset? Based on the IDs in the MIMIC_III_ECG_filenames.txt file, it appears like this may be the case. If so, I think that’s a potential issue. As the authors mention in line 177, mHealth applications usually use lead I (e.g., Apple watch, AliveCor use lead I). Some clarification would be helpful.

**Documentation:**

Generally, the documentation is good. It would be great if the authors could also share the preprocessing code used to generate the dataset.

**Ethics:**

It would be nice if the authors used the MIMIC matched waveform database so that demographic information could be used in the evaluation (or even model development). It’s possible that imputation performance differs based on age, sex, race, comorbidities etc. and without including this demographic information it’s impossible to know if this bias exists.

**Relation To Prior Work:**

Yes, the authors make it clear how their work differs from prior work.

**Summary And Contributions:**

The authors present a dataset for benchmarking the performance of ECG and PPG waveform imputation methods by leveraging publicly available ICU waveform data and using realistic missingness patterns to create simulated gaps to be imputed. While I believe that some of the baseline model choices don’t make sense, overall, the paper is well written and timely. They share the waveform data along with missingness masks, along with train/validation/test splits so that others can easily compare imputation methods to baseline model performance. Several baseline models from the literature were implemented and tested, along with a novel transformer model to serve as an additional baseline.

---

> ### Author Response · Authors · 2022-08-12
> **Response to Reviewer 7UND (1/3)**
>
> Thank you for the kind words and great suggestions. We have implemented the ideas you have suggested (i.e. simple FFT imputer and confidence intervals) and added our discussions into the revised paper. We address each of your questions in detail below. Please let us know if you have any further questions or need any additional clarifications.
>
> **Weaknesses**
>
> - "**mean and linear imputation [do not] necessarily make sense as baselines":** The rationale for including mean and linear imputation is that they are commonly used in mHealth applications (Le et al., 2018; Dong et al., 2019) and provide a way to baseline the accuracy of more sophisticated methods, which we have clarified in our revised text in Lines 225-229. Note for example that the Extended Loss section of Fig. 3 demonstrates that many of the more complex imputation methods have minimal performance benefits over a simple mean filling imputation.
> - " **use a simple FFT-based baseline":** Great idea.We followed this suggestion and used the FFT imputation procedure described by Rahman et al., 2015 as an additional baseline. We include results within our revised paper in  Figure 1, Figure 3, Table 2, and Table A3. In the ECG Heartbeat detection task, we found that the FFT imputer achieves comparatively low performance compared to BDC, but had the second highest F1 score among all methods. Furthermore, it is the only other baseline method that is able to reconstruct signals with realistic ECG morphology in the ECG MIMIC dataset, as seen in Figure 1. In the PPG Heartbeat detection task, we similarly find that the use of a frequency-based representation is effective, yielding a similar F1 score as in the ECG case. However, for Cardiac Classification in Figure 3, it seems that the FFT-based imputer's reconstruction is too noisy for effective classification, with its downstream cardiac classification being amongst the lowest performing imputation models. We will add additional details and results in Section 5 and the appendix in the final version.
> - "**[Does] not utilize the MIMIC matched waveform database"**: Interesting suggestion. Two comments: 1) Because we curated the entire MIMIC-III Waveform database (see Appendix A2), PPG signals from 6,532 patients and ECG signals from 12,677 patients _from the matched subset_ are included in our curated dataset. As a consequence, this work does enable future research to incorporate information from clinical data as covariates. 2) We chose not to utilize such covariates in the present work, due to our goal of focusing on the pulsative aspects of mHealth signals and in order to cast the broadest possible net and tackle the most common case of sensor-based imputation. In practice, there is no uniform set of covariates that will be available for all mHealth studies, as different sets of covariates might be available for different studies due to variations in IRBs and privacy concerns.

---

> > ### Author Response · Authors · 2022-08-12
> > **Response to Reviewer 7UND (2/3)**
> >
> > **Correctness:**
> >
> > - **"What do the authors mean by 'pooled all channels together'"**: This means that we take each of the simultaneously-recorded channels from clinical ECG data and add them separately to the dataset as individual univariate time series. By separating out the channels, we are effectively creating a "union of leads" dataset, thereby modifying the multivariate time-series into multiple univariate time-series. Now, we'd like to clarify why we used this "union of leads" approach for MIMIC, but not for PTB-XL:
> >   - **Union of Leads Dataset for Imputation in MIMIC:** MIMIC's unstructured nature with variable and imprecise lead placements per recording (Gow, 2021) lends itself to a union of leads approach, which has two main strengths, which we detail below. The first strength is that the inclusion of different leads creates a more interesting imputation problem because methods must learn to borrow information across time to capture morphology due to the distinctively different shapes present in each lead. The second strength is that it enables generalizability in the mHealth space because although Lead I is the most popular, other works have experimented with other leads. For example, Gupta, 2020 captures aVF; Lai et al., 2020 captures non-standardized leads; and Sprenger et al., 2022 captures V1-6.
> >   - **Lead I Only Dataset for PTB-XL:** The inclusion of multiple leads is not used in this case due to PTB-XL's dataset design (Wagner et al., 2020; Strodthoff et al., 2020). We need to adapt the 12-lead multichannel ECG classifier originally proposed for PTB-XL (Wagner et al., 2020)  to the univariate setting in creating a downstream task. Since our goal is to assess imputation performance on classification and the domain shifts associated with using different leads would be a confounding factor, we align on using a single lead, Lead I, in designing the experiment. Thus, we train the cardiac classifier once on non-ablated Lead I data, and then evaluate the trained classifier on each imputation model's imputations on separate, held-out Lead I data with missingness, in order to directly evaluate the impact of imputation on classification performance. For the sake of completeness, we will include a union of leads experiment on Cardiac Classification in the final version of the paper.
> >
> > **Documentation:**
> >
> > - **"share the preprocessing code"**:  Good catch. We have included it in our updated repository.
> >
> > **Additional Feedback:**
> >
> > - **"confidence intervals"**: Thank you for this suggestion, we have added this analysis to the paper. Following the approach of (Strodthoff et al., 2020; Zhou et al., 2019), we provide 95% confidence intervals by bootstrapping test samples with 1,000 iterations with 1,000 set seeds. These new results can be found in Table 2 of our revised paper, which we show below, and in Table A3 in the appendix.
> > - **Clarity on Appendix A1 and** **"reconstruction of morphology is important":** To clarify, we are in agreement that modeling signal morphology is important for our pulsative imputation task. Our focus on reconstructing signal morphology and measuring its impact on downstream tasks is aligned with the reviewer's comments. We have fixed this wording in the revised supplemental. The suggestions regarding LVET and T-wave identification are exciting directions for future work, and we have added this discussion into the Discussion Section.
> > - **Suggestions** : We have added the # of patients into the respective sections and included folders (e.g. root/data/mimic\_ecg/mimic\_ecg\_train\_separated/)  in our data dropbox folder with separated, individual waveforms. The suggestion of additional data for a personalized ML model approach is intriguing, and we added discussion of this approach in our Future Work section in lines 316-318.

---

> > > ### Author Response · Authors · 2022-08-12
> > > **Response to Reviewer 7UND (3/3)**
> > >
> > > **References:**
> > >
> > > Le, T. D., Beuran, R., & Tan, Y. (2018, November). Comparison of the most influential missing data imputation algorithms for healthcare. In 2018 10th International Conference on Knowledge and Systems Engineering (KSE) (pp. 247-251). IEEE.
> > >
> > > Dong, X., Chen, C., Geng, Q., Cao, Z., Chen, X., Lin, J., ... & Zhang, X. D. (2019). An improved method of handling missing values in the analysis of sample entropy for continuous monitoring of physiological signals. Entropy, 21(3), 274.
> > >
> > > Rahman, S. A., Huang, Y., Claassen, J., Heintzman, N., & Kleinberg, S. (2015). Combining Fourier and lagged k-nearest neighbor imputation for biomedical time series data. Journal of biomedical informatics, 58, 198-207.
> > >
> > > Gupta, A. (2020). Cardiosense. Retrieved August 6, 2022, from https://cardiosense.com/
> > >
> > > Lai, D., Bu, Y., Su, Y., Zhang, X., & Ma, C. S. (2020). Non-standardized patch-based ECG lead together with deep learning based algorithm for automatic screening of atrial fibrillation. IEEE journal of biomedical and health informatics, 24(6), 1569-1578.
> > >
> > > Sprenger, N., Sepehri Shamloo, A., Schäfer, J., Burkhardt, S., Mouratis, K., Hindricks, G., ... & Arya, A. (2022). Feasibility and Reliability of Smartwatch to Obtain Precordial Lead Electrocardiogram Recordings. Sensors, 22(3), 1217.
> > >
> > > Gow, B. (2021, April 30). Ambiguity in ECG lead label "V" and "MCL" in Mimic-III waveform dataset · issue #907 · MIT-LCP/Mimic-Code. MIT-LCP/Mimic-Code. Retrieved August 12, 2022, from https://github.com/MIT-LCP/mimic-code/issues/907
> > >
> > > Wagner, P., Strodthoff, N., Bousseljot, R. D., Kreiseler, D., Lunze, F. I., Samek, W., & Schaeffter, T. (2020). PTB-XL, a large publicly available electrocardiography dataset. _Scientific data_, _7_(1), 1-15.
> > >
> > > Strodthoff, N., Wagner, P., Schaeffter, T., & Samek, W. (2020). Deep learning for ECG analysis: Benchmarks and insights from PTB-XL. _IEEE Journal of Biomedical and Health Informatics_, _25_(5), 1519-1528.
> > >
> > > Zhou, N., Jiang, Y., Bergquist, T. R., Lee, A. J., Kacsoh, B. Z., Crocker, A. W., ... & Salakoski, T. (2019). The CAFA challenge reports improved protein function prediction and new functional annotations for hundreds of genes through experimental screens. _Genome biology_, _20_(1), 1-23.

---

### Review · Ethics_Reviewer_u48r · 2022-08-22

**Recommendation:** 1

**Ethics Documentation:**

The dataset is sufficiently documented.

**Ethics Review:**

The dataset is composed of existing publicly available medical signal datasets and missigness patterns. The appendix does a good job at listing all the component datasets, and their usage rules.

I only have some concerns with the wording in Appendix A7:

> The data does not contain personally identifiable information nor offensive content.

It is not clear whether this comment refers to the full dataset or the missingness patterns. If it refers to the full dataset, I would argue that this is incorrect: The signals such as ECG are personal data (see, e.g., [1]), thus are personally identifiable even if with low likelihood. I recommend removing this statement if it it is intended to refer to the whole dataset. Instead the details could explicitly say, e.g., that the information is personal data with identifiers removed, yet it is publicly available for research purposes and its redistribution is in public interest.

> We, the authors bear all responsibility in case of violation of rights, etc., and confirmation of the data license.
It would be better if the text clarified what exactly is meant here, i.e., what responsibility and which rights.

[1] https://pubmed.ncbi.nlm.nih.gov/26272456/

---

> ### Author Response · Authors · 2022-08-24
> **Response to Ethics Reviewer u48r (1/1)**
>
> We thank Ethics Reviewer u48r for their positive comments and assessment of our work. We will address their wording concerns below.
> * **Clarification on Personally Identifiable Information:** Thank you for bringing this to our attention, and we have modified the wording in lines 202-203 in our revised Appendix with your suggestion.
> * **Clarification on Author Statement on Bearing Responsibility:** We have further clarified this statement in lines 206-207 in our revised Appendix.

---

### Author Response · Authors · 2022-08-12
**Response to all Reviewers**

We want to thank all of the reviewers for their positive comments and great suggestions for improving the paper. We added 12 more experiments to benchmark a new FFT-based imputation baseline, as well as re-running all experiments to include confidence intervals. This has resulted in updates in Figure 1 and 3, Tables 2 and Table A3, and in the Benchmarks and Results Sections. In addition, the Introduction, Related Work, PulseImpute Challenge Description, and Discussion sections have all been updated to reflect the reviewer feedback. To enhance reproducibility, we have further included curation code in our GitHub repo. Please let us know if you all have any further questions, and we would be more than happy to address them.

---

### Meta-Review · Area_Chair_Wage · 2022-09-13

**Recommendation:** Accept
**Confidence:** 4

**Metareview:**

This paper develops a new benchmark for missing data imputation in pulsative signals like ECG using realistic missingness models. I expect such a dataset to drive important developments in this understudied area; indeed, the authors show that standard SOTA methods fail. The reviewers' enthusiasm makes this paper a clear accept.

---

### Decision · Program_Chairs · 2022-09-16

Accept